# Heritage Reconstruction Planning, Sustainability Dimensions, and the Case of the Khaz'al Diwan in Kuwait

**Reyhan Sabri** [1,*] **, Haşim Altan** [2,*] **, Danah AlGhareeb** [3] **and Noora Alkhaja** [4]

1   Architectural Engineering Department, College of Engineering, University of Sharjah, Sharjah 27272, UAE
2   Department of Architecture, Faculty of Design, Arkin University of Creative Arts and Design, Kyrenia 99300, Cyprus
3   Ministry of Public Works, Kuwait City 65000, Kuwait; alghareebdanah@gmail.com
4   Architectural Engineer, Dubai, UAE; noura.alkhaja@gmail.com
*   Correspondence: rsabri@sharjah.ac.ae (R.S.); hasim.altan@arucad.edu.tr (H.A.)

**Abstract:** Although attempts for formulating sustainable approaches in heritage management have been ongoing since the 1980s, sustainability dimensions in the context of 'reconstruction' have remained an unexplored research area. By investigating the case of the ruined Khaz'al Diwan in Kuwait, an architectural heritage site in the United Nations Educational, Scientific and Cultural Organization (UNESCO) World Heritage (WH) Tentative List, we explore and compare the roles of the 'cultural continuity' and 'environmental protection' pillars of sustainability in reconstruction planning. By employing rapid ethnographic surveying and case study methods, we first investigate the approach to 'cultural continuity' from the State's stance and through local community perceptions. Albeit with nuances, the surveying revealed a preference for historicist reconstruction. However, the Khaz'al Diwan, like most of the heritage structures in the Gulf region, was originally constructed with coral stone, which is now protected under environmental laws. How feasible is the use of replacement materials in terms of sustainability perspectives that is also acceptable from heritage perspectives? Considering the high cooling loads required in this climatic region, we prioritized the energy performance of the construction materials of the external walls and the roof. Computer simulations based on scenarios testing same-type and replacement construction materials revealed how the latter could be considered as an alternative in a historicist reconstruction. The discussion revolves around the environmental and cultural parameters that are instrumental in reconstruction planning. This ultimately highlights how reconstruction policies must be shaped to redefine the role and scope of material authenticity to accommodate the local environmental and cultural realities in the wider Gulf region and Middle Eastern context.

**Keywords:** architectural heritage; historicist reconstruction; cultural continuity; environmental sustainability; energy efficiency

## 1. Introduction

'Reconstruction' is perhaps both philosophically and practically the most challenging physical intervention process in architectural heritage. As reconstruction is the process of 'returning a place, which is incomplete through damage or alteration, to an earlier state of the fabric' [1], it has been widely assumed in Eurocentric heritage discourse as a physical disruption to the authenticity of the physical fabric of any cultural property [2]. Hence, documents aiming for international conservation standards include rather rigid definitions for the conditions that justify reconstruction, cautiously restricting it to exceptional circumstances, such as destruction due to wars or natural catastrophes.

The Declaration of Dresden on the Reconstruction of Monuments Destroyed by War (1982), which is considered a pioneering document in the field of post-war recovery of destroyed cultural properties, justifies reconstruction as 'Arising from the legitimate desire of peoples to restore damaged monuments as completely as possible to their national significance' [3]. Recently, the Warsaw Recommendation on Recovery and Reconstruction of Cultural Heritage (2018) endorsed the reconstruction of heritage sites as a tool for the recovery of society in the aftermath of conflicts and disasters [4]. However, as Bold [5] noted, while the need to reconstruct after war or natural disaster is clear, that requirement does not prescribe how it should be done.

The Venice Charter of 1964, which was the first influential document to codify internationally accepted architectural conservation standards, prioritized materialist authenticity, and ruled out all reconstruction work 'a priori' [6]. Many key international documents, arising after, adopted this philosophy and restricted reconstruction to exceptional circumstances and instructed its implementation in a highly meticulous scientific methodology, based on original documents, hence, privileging the authenticity of physical form, fabric, and setting [7–9]. This physical fabric-based authenticity approach was debated in the historic Nara Document on Authenticity (1994) in favor of the inclusion of diverse cultural heritage forms and processes [10].

The Nara+20 document of 2014 [11], which was issued with an aim to further consolidate and develop the achievements and concepts of the earlier Nara document, acknowledged 'authenticity' through a constructivist approach as a quality that is culturally divergent [12]. Despite the expanding meanings of 'authenticity' and the growing international acceptance for a more flexible approach in reconstruction planning [5], UNESCO WH Convention adopted, since its inception in 1972, the Venice Charter's materialist approach and dogmatic tone against the 'reconstruction' of architectural properties, which is still the fundamental principle justifying the Outstanding Universal Value (OUV) in WH Sites [13]. Albeit, this can be extremely problematic in post-war contexts.

Researchers have argued that attempts to follow authenticity-oriented materialist heritage guidelines could have negative consequences in a post-war/disaster society not only due to limitations in material and human resources, but also as they result in lengthy debates and disagreements on the style and quality of reconstructions [14,15]. In other cases, haphazardly planned rapid reconstructions immediately after disasters have resulted in incoherent and controversial treatment of historic fabric [16].

Other authors argued that there is no sufficient information to determine what are appropriate or inappropriate methods in reconstruction planning in general [13], as the international guidelines barely address the practice-related issues and technicalities [17], and each situation requires a context-tailored approach [18,19]. According to Bold [5], debates and practice tend to polarize two alternatives, which include replicating the original appearances and materials in a historicist style for reasons of continuity and identity; or rebuilding in a contemporary style, signifying a new beginning. While each case is determined within their local practicalities, reconstructions practiced for the WH Sites typically follow the former [2].

On the other hand, the role of cultural heritage in sustainable development has started to receive increasing recognition since Nara+20 [11]. The Warsaw Recommendation [4] aligned the restoration of cultural assets with the principles of sustainable development and the 'build back better' approach. This is especially important for the Middle East and North Africa (MENA) region, which have experienced heritage destruction due to armed conflicts since the last decades of the 20th century [18,20]. Studies have indicated various aspects of the post-conflict reconstruction of cultural heritage from approaches and methods for recording, inventory formation, and condition assessment of damaged structures and identifying priorities in their reconstruction [21], to the political underpinnings of reconstructions and the importance of the engagement of local communities in the reconstruction processes [22,23].

However, research on the sustainability dimensions in heritage reconstructions in the region and elsewhere is very thin. Endeavoring to widen the knowledge in this avenue, in this paper, we investigated the case of the Khaz'al Diwan, a war-destroyed heritage building in Kuwait.

The investigation considered two pillars of sustainability, namely cultural continuity and environmental protection, and explored their role in reconstruction planning. 'Cultural continuity' is addressed as the main pillar underpinning architectural conservation processes, and 'environmental protection' is addressed in relation to the use of construction materials/layers and their impact on the environment. What is the official mission, expert opinions, and community perceptions regarding the future of the site? How is the environmental sustainability impacted by using same-type materials and replacement materials in reconstruction? Could using replacement materials be an alternative solution for the WH Sites with limitations in traditional material resources, and how will the issue of authenticity will be negotiated in this case?

In the first phase of this research, we looked at the cultural dimensions by employing rapid ethnographic surveying and content analysis of primary sources. Via questionnaires and interviews, we investigated how the local community and heritage professionals perceive the reconstruction of the historic site of the Khaz'al Diwan. The information regarding the preparations for planning, strategies, and sustainable development goals was retrieved from official documents and interviews with the heritage professionals in charge with the site. In the Second phase, we looked at the environmental impact of the use of materials, specifically focusing on the energy saving issue, which, in Kuwait, due to its climatic conditions, is mostly about the cooling load.

We investigated this in two scenarios: (1) reconstruction with existing original/traditional and same-type materials to fulfil the authenticity concerns and (2) reconstruction with replacement materials to make it more sustainable in terms of energy efficiency if doing so is sensible and feasible. The analysis of the findings revealed conflicting, converging, and deviating points between the pillars of sustainability and materialist conservation directives, and also delineated a set of criteria to be considered instrumental for the development of sustainable reconstruction planning and policies in the wider Gulf region.

## 2. The Khaz'al Diwan in the Historical and Cultural Context: A Residential Property, A National Museum, A Historic Monument and A War-Ruin

The Khaz'al palaces, namely the Qasr (residential palace) and the Diwan (guest house), were built in the beginning of the 20th century at the Dasman area in Kuwait City by Sheikh Khaz'al Bin Mirdau, the then Arab ruler of Khuzistan [24]. Finished in 1916, both were large and elaborate two story structures, in a stark contrast to Kuwait's primarily single story, austere architecture [25–27]. In the early 1940s, the Qasr was acquired by Ahmed Mohammad al-Ghanim, and the Diwan by Sheikh Abdullah Jabir al-Sabah, both to be used as family residences [25]. The focus of the present paper is the Diwan, which also is known in local history as the Sheikh Abdullah Al-Jabir Palace.

Unlike the Qasr, which was designed with the traditional courtyard layout, the Diwan featured a hall layout with four cylindrical towers punctuating its corners (Figure 1). Research has argued it to be the first building in the country, and perhaps in the region, with external landscaping but no internal courtyard [26,28]. Accordingly, the structure contained a basement, ground floor, and first floor, with each floor having a separate entrance. Each floor had six guest rooms distributed linearly along a central corridor.

External verandas extended in all elevations, demonstrating the addition of architectural imports to Kuwait's previously traditionally built environments. Pointed lobbed arches fronted the porches on the front and back elevations, and along the internal corridors, indicating Indian connections. The balconies and the wide eaves running along the building were supported on wooden columns. While the plan layout and architectural style pointed to external influences, the building was erected in the traditional construction methods: walls in coral stone masonry, plastered with gypsum, and the roof structure consisting of mangrove poles, topped by wooden boards and a layer of mud.

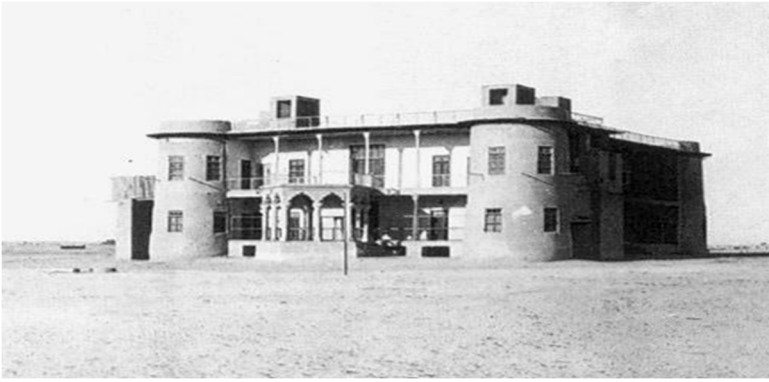

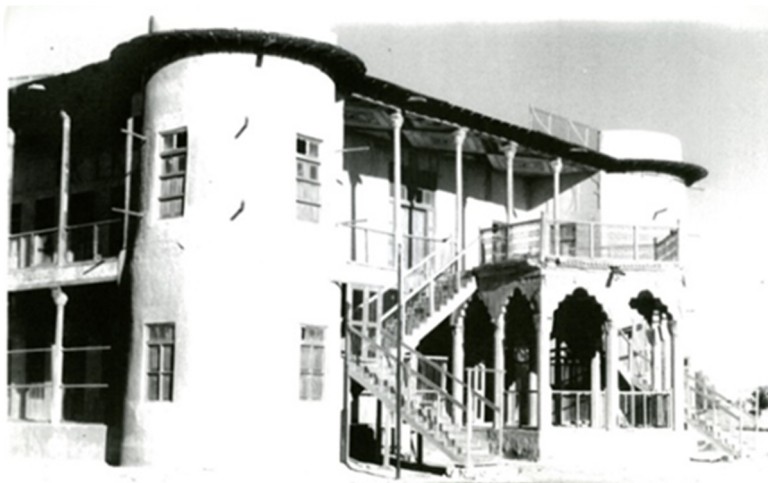

**Figure 1.** Front view (top) and back view (bottom) of the Khaz'al Diwan in ca. 1927, in its original state as built by Sheikh Khaz'al (source: National Council for Culture, Arts & Letters archives).

In the subsequent stage, Sheikh Abdullah Al-Jabir Al-Sabah added a fence surrounding the site, and he built four villas and two service buildings in the modern style for his growing family. However, when, in 1954, Kuwait witnessed a heavy rainfall that caused damage to the Diwan, the family evacuated from the building. Soon after that, Sheikh Abdullah Al-Jabir, then a leading political figure in the cultural and educational issues of Kuwait, rented the Diwan to the state for housing the first national museum of Kuwait [25,28]. The National Museum, which occupied the ground floor and the basement of the Diwan, opened its doors at the end of 1957 [24]. The first floor was inaccessible due to the fragility of the flooring [28]. In 1976, the museum was evacuated due to its fragile condition, and the collections were moved to newly built premises. The Diwan became a bachelor tenement afterwards and through the 1980s, causing further damage to its physical fabric [24]. In a Kuwait Historical Preservation Study prepared in 1988, it was described as having a 'High ++ Preliminary Preservation Rating' [28].

The Khaz'al Diwan has been a Grade-1 historic monument since the establishment of the Kuwait Heritage Building Register in 1997 [24]. As per the Kuwait Antiquities Law of 1994 (the amended Law of 1960), any physical intervention on the inscribed buildings/sites has to be done strictly under the regulations and stipulations determined by the National Council for Culture, Arts & Letters (NCCAL; hereafter Council), established in 1973.

Sadly, the Diwan was shelled during the Iraqi invasion in 1990–1991, leaving it in ruins [29]. Only the cylindrical corner towers and some partial walls survived the war (Figure 2). The site was acquired by the Council in 2008, and it has been on the UNESCO WH Tentative List since 2015. The remaining fragile walls of the Diwan, which had structural weaknesses even before the war destruction, have further decayed as no consolidation work has been performed since the war [28].

Recently, after a rainstorm in November 2018, one of the four cylindrical towers collapsed, and there are structural cracks on the remaining parts of the other three towers.

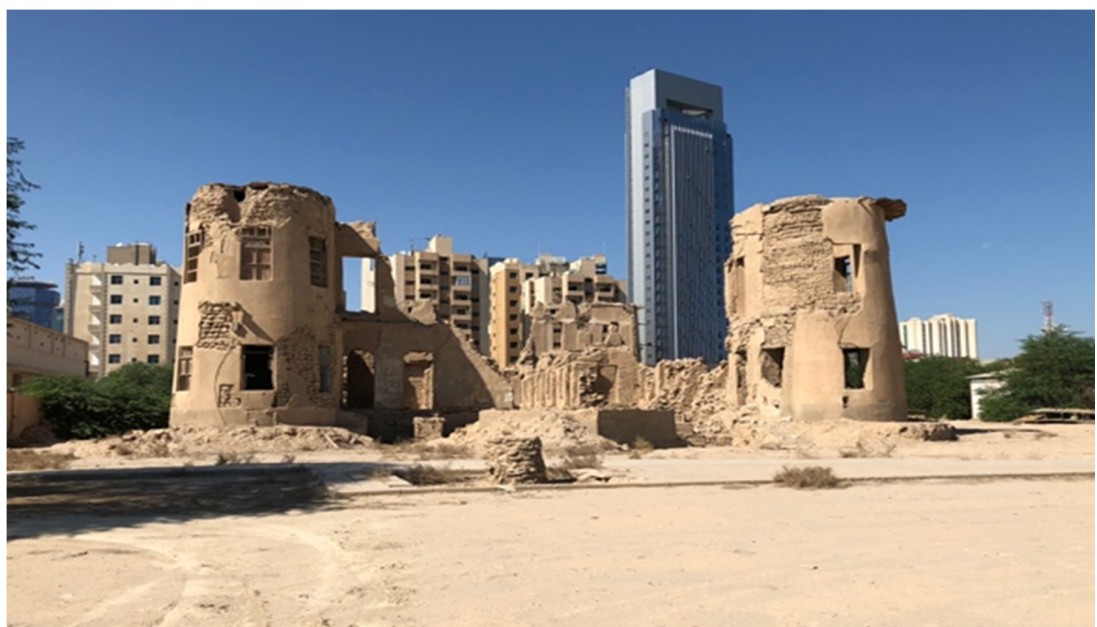

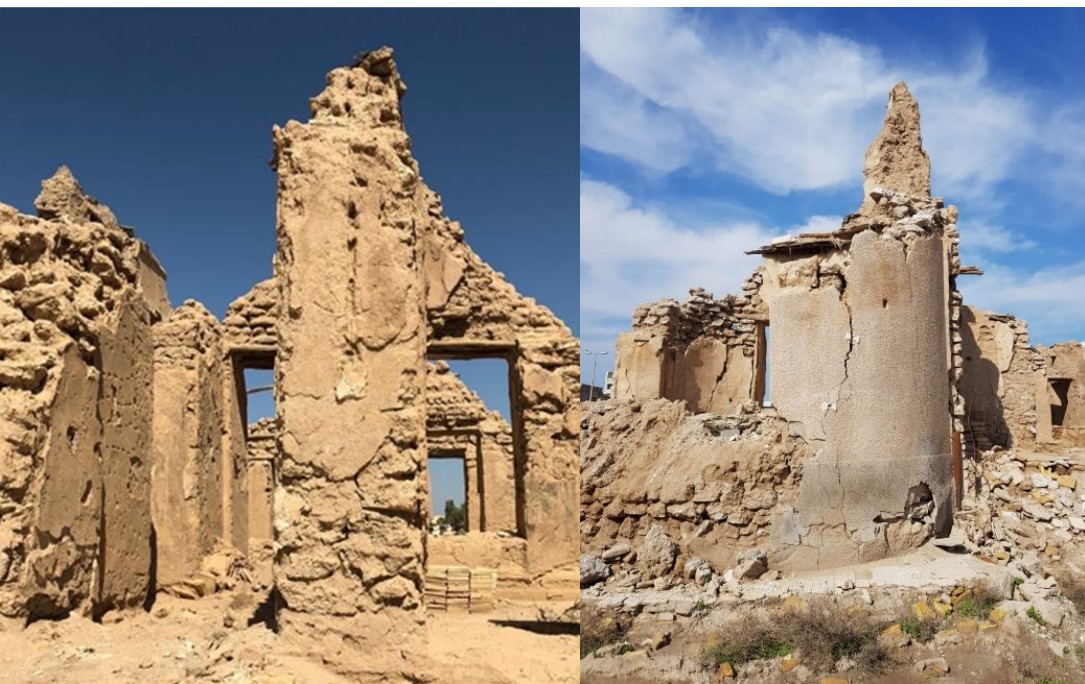

**Figure 2.** Recent views from the remains of the Khaz'al Diwan, Kuwait City.

## 3. Contemplating A Future for the Historic Site of the Khaz'al Diwan in the Light of Authorized Heritage Discourses and Sustainability Perspectives

In this section, we analyze the cultural dimensions underpinning the planning for the future of the Khaz'al Diwan through the exploration of the Council's (and hence the State's) approach and the community perceptions. We first address the Council's approach through interviews with heritage experts and the content analysis of two key documents accessible to the public: a booklet [24], published by the Council in 2010 (hereafter NCCAL 2010) and a Terms of Reference [28], prepared

in 2016 (hereafter NCCAL 2016). The community perceptions regarding the Diwan were analyzed through responses to a questionnaire survey that was distributed to 200 Kuwaitis.

After acquiring the Diwan in 2008, the Council initiated a surveying and documentation process to collect information on the historic layers on the fabric and architectural characteristics [24], echoing the directives of the Venice Charter of 1964 [6]. Accordingly, following the initial investigation, and clearing of the debris, a detailed three-dimensional/photogrammetrical survey was planned, with the aim to produce measured drawings, digital modelling, a precise record of the construction techniques/materials, and the successive building evolution stages. The Council determined that 'the restorations will respect the Venice Charter, necessitating that traditional materials be used, and traditional building techniques applied' [24].

The Council explained in NCCAL 2010 that the collection of all recyclable materials, such as mud bricks, loose clay, and coral stones, would be employed during the restoration phase and that laboratory analyses would be conducted on the materials [24]. The state of the ruins and the endeavors for collecting as much information as possible regarding the architectural fabric indicated that what was defined in the booklet as 'restoration' was going to be 'reconstruction'. The word 'reconstruction' or 'rebuilding' was neither termed in NCCAL 2010 nor in the subsequent NCCAL 2016, as this was a process abhorred by the Venice Charter [6].

Instead, the Council put an emphasis on the following of the directives of the Venice Charter [6] which would ultimately result in 'an exemplary work, which will set the standard for heritage preservation and restoration in the State of Kuwait and in the Gulf' [24]. The documentation continued, resulting in a detailed report on the materials and structure in 2011, prepared by Dr. Ahmed AlShamali's Engineering Consultancy [30]. In 2013, the building was 3D scanned and documented as part of an archaeological survey and excavation by a Kuwaiti-Georgian expedition team. Finally, architectural drawings and 3D modelling sets were created, reflecting the building's original design in 1916, and the second phase with renovations during its conversion into a Museum in 1957 (clause 4.1.6 in NCCAL 2016).

However, the lack of a deadline for finalizing the planning and starting the implementation has marred this ambitious mission. The Council announced in 2010 that 'the project will take several years to complete' [24]. This is not an uncommon situation, and as Dewi [15] has argued for other contexts after Al-Nammari and Lindell [14], it often happens 'because of the debates and disagreements over material damage estimates, the cost of repairs and the level of expected quality of the re-building.' With the mission to register the historic site of Khaz'al Diwan on the WH List, which required meticulous planning as well as complying with the rigid stipulations of WH Convention, the decision-making process became a prolonged journey. The war-ruined structure was left bare under Kuwait's rainstorms, pending the finalization of the decision-making and planning processes.

By 2015, when it was registered in the WH Tentative Lists, the Diwan was 'in a very bad state of conservation. The only remnants of the original building amount to the corner towers, the front entrance, and some walls. The floorings have been obscured by debris of collapsing walls and roofs. All interior and exterior doors and windows have been lost, except for a few window frames still surviving on the outer towers, albeit in very poor condition' [27].

The Terms of Reference document, prepared in 2016, specified that a 'restoration' work was planned on the site of the Diwan, and aimed 'to reflect its original status when it was built in 1916' [28]. Like the 2010 document, the 2016 document echoed the Venice Charter, specifying that 'Original building material and methods should be respected and any additions and equipment required should not interfere with the original design and should be minimal' (article 4.1.6 in NCCAL 2016). Again, despite identifying the process as 'restoration', the descriptions in the document reveal that this would be a historicist reconstruction, in which the Khaz'al Diwan was intended to be rebuilt on the same site, in the same architectural, volumetric, and construction characteristics, and using the existing original materials and same-type ones.

The 2016 document emphasized that international expertise, such as that of International Council on Monuments and Sites (ICOMOS), would be followed with the aim to achieve UNESCO WH requirements (clause 5.1.2 in NCCAL 2016) with a conservation management plan focusing on the protection of the OUV (clause 5.3.1.2 in NCCAL 2016). Sustainable development goals were also considered in the light of the WH criteria. Clause 5.5.8 of the document states that the proposed use of the site should be 'environmentally and culturally sustainable', in conformity with the WH criteria [28].

While the clause does not define 'environmental and cultural sustainability', it sets forth a few generic goals. Accordingly, 'cultural sustainability,' is associated with contributions to the 'quality of life of the communities concerned', and 'environmental sustainability' is associated with having 'a minimum environmental impact in the design, construction and operation phases.' It is also highlighted in the same clause that such sustainable use or any other change should not impact adversely on the OUV of the site. Put differently, the authenticity of the historical fabric should not be compromised by the set sustainability goals. As it has also been noted by Khalaf [13], one of the three pillars of OUV is 'the conditions of authenticity and integrity', which indicates that if the physical fabric is heavily damaged and reconstruction is required, the OUV status is considered jeopardized if not altogether lost.

How important is the material authenticity of the Diwan for the lay person? Our questionnaire was responded by 200 Kuwaitis. While 15% indicated no interest in the site's preservation, 85% of respondents (169 respondents) opined that the site of the Khaz'al Diwan should be preserved. We found that 46% of this sample group indicated that the structure should be preserved as ruins for memorializing the destruction of war, but also because they are 'nice', 'interesting', and 'educational'. On the other hand, 54% of 169 respondents favored the reconstruction of the Diwan. A total of 68% of them justified their opinion as 'the evidence of Kuwait's early 20th century architecture', as an 'important historic layer in Kuwait's urban fabric', as the 'first museum in Kuwait,' and as 'evidence of the pre-oil period.'

Put differently, like the State, the community indicated a preference for the historicist reconstruction for reasons of cultural continuity. On the other hand, while 27% of the total respondents disagreed on any contemporary development on the historic site of Khaz'al Diwan, 73% noted that the site would be 'lively' and 'more attractive' with contemporary touches, as 'heritage sites should not be static', but be 'dynamically evolving'. Although more extensive ethnographic surveying needs to be conducted for a wider opinion, this rapid ethnographic surveying indicates how the community members do not perceive the 'cultural continuity' in freezing the monument, but in linking it to the present and contributing to its evolution.

Khalaf [13] rightfully argued after Smith [31] that heritage is 'a process in which cultural and social values are rewritten and redefined for the needs of the present.' Therefore, as Khalaf [13] argues, perhaps it is time to push into the background the importance accorded to the physical fabric of historic architecture and bring to the forefront the meanings and values associated with reconstruction. Other authors argued that a key step toward achieving this goal is through inclusive heritage management and, hence, by involving the community members as stakeholders in the decision-making process [12,32]. In this case, the community members' interests for the protection of the Khaz'al Diwan's site as a museum, for its reconstruction in a historicist fashion as an expression of identity and cultural continuity, and for its evolution witnessing the present are key local guidance points.

This brings us to the non-renewability of the material fabric—an overarching criterion for the earning and protection of the OUV status—and the situations in which the supply of same-type materials have negative consequences on the environment. Until the oil era, the use of coral stones in vernacular architecture, usually mixed with sea stones, and bound together with mud mortar derived from seashell stone was widely practiced in the Gulf region [33]. However, in Kuwait [34], and elsewhere in the Gulf region [35], the coral stone resources (coral reefs and sea-beds) are now protected under environmental laws and the protection of coastal environments is considered among sustainable development goals.

When the requirement for coral stones in architectural conservation works in the region is in low quantities, they are typically sourced from demolished vernacular buildings [36]. However, considering the massive scale of the Khaz'al Diwan with its wall thicknesses ranging from 80–100 cm, a massive amount of coral stones would be required. Although the Council has collected the materials from the demolished structure, aiming to reuse them in the reconstruction, most of the coral stones have already lost their structural properties.

As such, there are environmental advantages in challenging the material authenticity in historicist reconstruction planning in the region. Could using replacement materials in historicist reconstructions in the region be a strong enough advantage to inspire re-shaping the authorized heritage discourses and, more importantly, the WH policy? We conducted simulations using same-type (authentic) and replacement material layers on the external walls and the roof of the Khaz'al Diwan with the aim to measure their efficacy on energy saving performance. This is explained in the next section.

## 4. Investigating Environmental Sustainability Dimensions in Reconstruction: Energy Efficiency Simulations with Same-Type (Authentic) and Replacement Materials

The provision of energy saving solutions and strategies for heritage buildings, for better environmental protection, has begun to receive wider attention, particularly since the 2010s [37–42]. Reducing energy consumption is an especially important challenge in the Gulf's climatic conditions. This section first presents the energy consumption performance of the Khaz'al Diwan as if reconstructed with same-type and same-dimension materials. Afterward, the energy performance with replacement materials was tested through building performance simulation software, and then both performances were analyzed.

The first scenario was based on the historicist reconstruction with same-type materials and same-dimension material layers; the external walls were modelled to include 800 mm thick coral stone masonry bound with mud-mortar and treated on the exterior and interior faces with 10 mm thick same-type (original-like) plaster. The windows and external doors were modelled in same-type wooden material. The roof was modelled according to the original materials and detailing, including mangrove poles and layer of wooden planks, topped by a layer of mud (Table 1, First Scenario).

The Second scenario was based on a historicist reconstruction with replacement construction materials for the external walls (made of double concrete block with a thermal insulation layer in between), treated on the exterior and interior faces with 20 mm same-type gypsum plaster. The finishing materials (plaster, decorations, doors, and windows) were modelled in same-type material, hence, preserving the historic aesthetics of the building. The roof was modelled with same-type materials and details, topped with water proofing (made of sprayed polyurethane foam, 30 mm) and a thick thermal insulation layer (EPS; expanded polystyrene, 350 mm) to reduce losses through the roof surfaces. The roof was topped by a protective layer, an asphalt reflective coating, 20 mm, which can be reflective in its nature, helping with the reduction of heat gains due to solar radiation through the roof surfaces (Table 1, Second Scenario).

With the use of computer modelling and energy performance simulation methods, the proposal for the historicist reconstruction with replacement construction materials was tested to improve the environmental sustainability dimensions—in this case energy efficiency—by reducing the cooling loads, which was demonstrated through the use of DesignBuilder software [43,44]. In such a case with replacement materials (Table 1, Second Scenario), the specific annual cooling demand was reduced by 21.45% compared to the historicist reconstruction building condition of the restored Khaz'al Diwan (Table 1, First Scenario).

Considering that the rate of use of the equipment, lights, and occupancy were the same in both scenarios, the most indicative aspect was the cooling energy demand and the energy efficiency of the historicist reconstruction with original/same-type versus the historicist reconstruction with replacement materials. The two building reconstruction scenarios are presented in Table 2.

**Table 1.** Comparative review of the two proposals: First Scenario (the historicist reconstruction with same-type materials and same-dimension material layers) and Second Scenario (the historicist reconstruction with replacement construction materials) including building envelopes, such as external walls, roof, and windows.

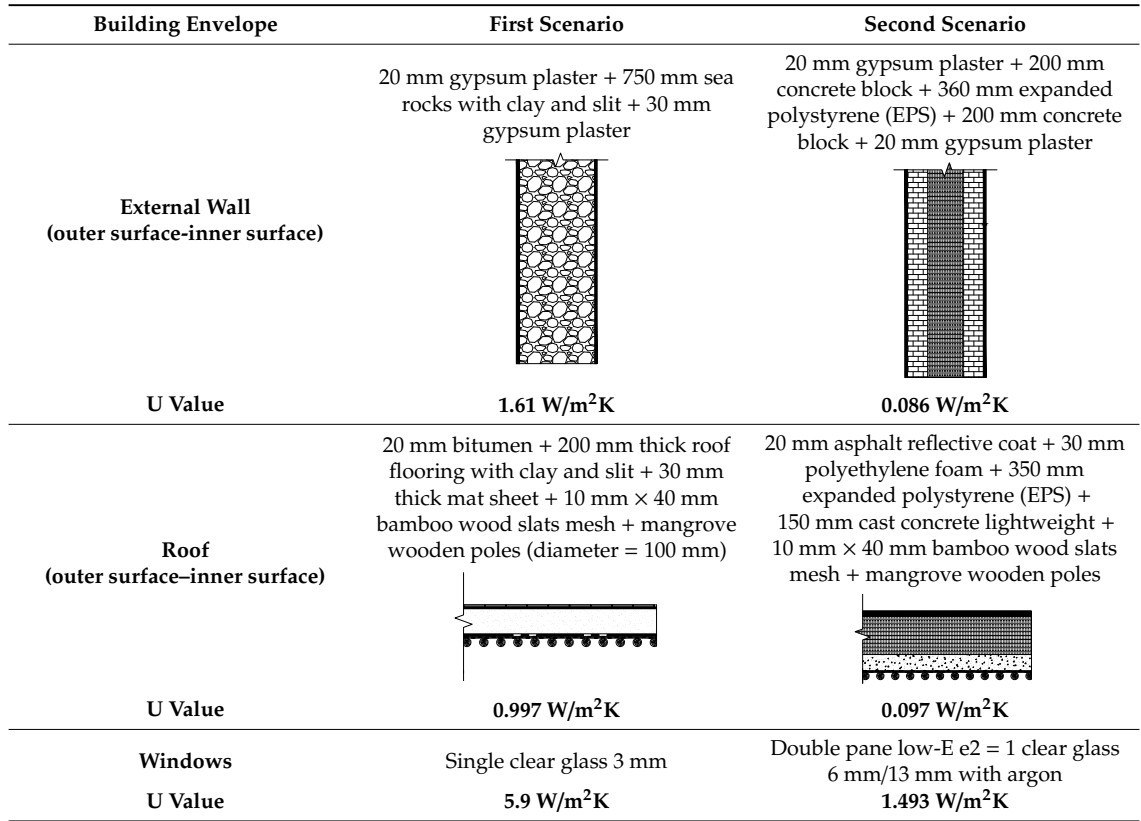

| Building Envelope | First Scenario | Second Scenario |
|---|---|---|
| **External Wall (outer surface-inner surface)** | 20 mm gypsum plaster + 750 mm sea rocks with clay and slit + 30 mm gypsum plaster | 20 mm gypsum plaster + 200 mm concrete block + 360 mm expanded polystyrene (EPS) + 200 mm concrete block + 20 mm gypsum plaster |
| **U Value** | **1.61 W/m²K** | **0.086 W/m²K** |
| **Roof (outer surface–inner surface)** | 20 mm bitumen + 200 mm thick roof flooring with clay and slit + 30 mm thick mat sheet + 10 mm × 40 mm bamboo wood slats mesh + mangrove wooden poles (diameter = 100 mm) | 20 mm asphalt reflective coat + 30 mm polyethylene foam + 350 mm expanded polystyrene (EPS) + 150 mm cast concrete lightweight + 10 mm × 40 mm bamboo wood slats mesh + mangrove wooden poles |
| **U Value** | **0.997 W/m²K** | **0.097 W/m²K** |
| **Windows** | Single clear glass 3 mm | Double pane low-E e2 = 1 clear glass 6 mm/13 mm with argon |
| **U Value** | **5.9 W/m²K** | **1.493 W/m²K** |

**Table 2.** The energy performance of the two separate scenarios (same-type vs. replacement).

| Energy Demand | First Scenario | Second Scenario |
|---|---|---|
| Room Electricity | 17.29 MWh | 17.29 MWh |
| Lighting | 34.57 MWh | 34.57 MWh |
| Cooling Load | 401.31 MWh | 315.22 MWh |
| **Total** | **453.17 MWh** | **367.08 MWh** |

Accordingly, it would be useful to see the monthly breakdown of the energy demand for each of the two scenarios (i.e., historic vs. replacement), as shown in Figure 3. The peak cooling demand was in the April–October period, having the lowest demand in December, January, and February. In the peak demand period, in July, the First Scenario (historic reconstruction) demonstrated a cooling energy demand of approximately 74,780 kWh and the Second Scenario (replacement reconstruction), on the other hand, showed a cooling energy demand of approximately 60,516 kWh, which was almost 18% less for the same period.

As expected, the cooling loads were higher during summer months and, for this reason, improvements on the building envelope would have a significant impact on the overall environmental sustainability as well as improving the energy efficiency during building operations. The carbon-dioxide emissions under both scenarios are presented in the following table (Table 3), and, accordingly, the carbon-dioxide emissions were much lower for the Second Scenario (replacement reconstruction) due to the more energy efficient building envelope/fabric.

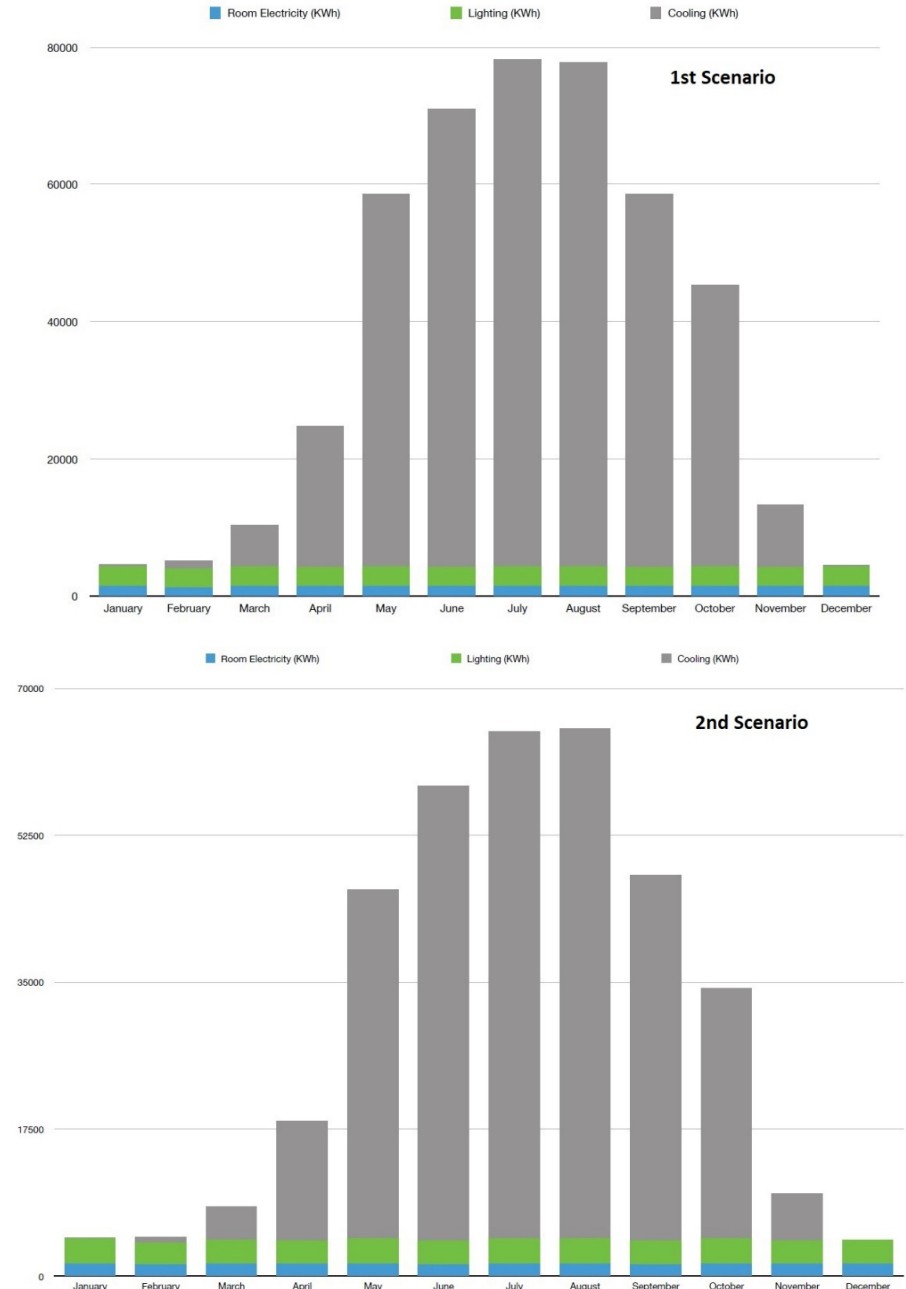

**Figure 3.** The monthly energy consumption of the two scenarios (source: DesignBuilder).

**Table 3.** Annual $CO_2$ emissions for both scenarios (historic vs. replacement).

| Emissions | First Scenario | Second Scenario |
|---|---|---|
| $CO_2$ (kg/m$^2$/year) | 174,000 | 141,000 |

A cost analysis was undertaken to demonstrate the total restoration and operating costs under both scenarios, and the below table (Table 4) presents the calculations made for each building component/element using construction materials as per Table 1. The costs are shown in US Dollars (USD) after conversion from Kuwaiti Dinar (KWD), where 1 KWD = 3.26 USD.

**Table 4.** Cost analysis of the two separate scenarios (same-type vs. replacement).

| Cost | First Scenario | Second Scenario |
|---|---|---|
| Wall (USD/m$^3$) | 130 | 71 |
| Roof (USD/m$^3$) | 110 | 114 |
| Window (USD/unit) | 88 (single glazed) | 114 (double glazed) |

As can be seen in Table 4, the costs vary depending on each building component/element; a higher cost could be justified with environmental benefits, such as a lower cooling demand due to a more energy efficient building envelope/fabric. As with the operational costs, the reconstructed building under both scenarios (historic vs. replacement) would have identical base electrical loads for lighting, etc. except for the cooling demand, which would differ due to more end-use with the less energy efficient building envelope/fabric as in the First Scenario.

## 5. Discussion: Challenging the Consensus in Authenticity-Based Reconstruction for A Better Way Forward

It has been almost four decades since the Declaration of Dresden justified that the desire to acknowledge architectural monuments both intellectually and politically were among the key reasons for initiating their reconstruction [3]. Armed conflicts, especially in the MENA region, have continued, causing ongoing severe destruction to historic structures. The reconstruction planning processes of many such structures has continued with debates on what is appropriate and what is inappropriate [18,20,23].

Orbaşlı [12] rightfully argued that 'debate on how much re-building, reconstruction and re-imagining is acceptable will continue to be played out under the influences of global competition, prevalent tourism trends, post-conflict and post-disaster recovery, and evolving community values.' The case of the Khaz'al Diwan demonstrates a key example for how global recognition through WH has become the State's mission during post-conflict recovery in Kuwait.

However, as other authors observed, authorized heritage discourses have been privileging the OUV, which places specific emphasis on material authenticity [13]. Consequently, the notion that other-type materials are detrimental for the integrity of heritage structures dominates the WH selection criteria for architectural heritage sites, creating constraints for reconstruction planning.

On the other hand, technical standards, or specifications to clarify the heritage reconstructions' potential in sustainable development have not yet been made clear. For instance, in the cases where the heritage structure is almost completely destroyed and the goal is to have a historicist reconstruction, this may require the resourcing of a substantial amount of same-type materials. However, the problems related to the sourcing of the same-type construction materials, which are under the protection of environmental laws, have been almost neglected. Similarly neglected is the energy performance analyses related to same-type and replacement materials in heritage reconstruction planning and the practicality of using replacement materials for a better environmental response. The present research investigated this topic through the case of the Khaz'al Diwan in Kuwait, which was shelled down and collapsed during the Iraqi invasion in 1990–1991.

As the findings of this research demonstrated, the State and the community members perceive a historicist reconstruction of the Khaz'al Diwan as a tool for their cultural continuity, identity, and resilience, which is similar to many cases in other geographies, reported for instance in Bold et al. [2]. At the same time, the main construction material, which is the coral stone resources in Kuwait—and elsewhere in the Gulf region—are protected by environmental protection laws. The analyzed documents and interviews revealed intentions for reusing the existing coral stone, mud brick, and floor finishes, as well as same-type materials salvaged from other demolished heritage structures. However, although this is a sustainable action, the reconstruction policy and future planning cannot be based solely on the availability of salvageable materials from other heritage structures. Similarly, while the emphasis at

present is placed on environmental sustainability and compliance with international standards, this mainly considers energy-efficient building services, and it does not extend to the building materials.

Aiming to widen the discussion on the re-shaping of the reconstruction policy within the context of environmental protection, we tested the energy performance levels of the Diwan for two different scenarios, based on use of same-type and replacement materials. These energy performance simulations are intended as a desktop study to help demonstrate the big picture for the reconstruction of such historical buildings in the future. As expected, the energy demand of the coral stone built Diwan with same-type materials and dimensions was considered higher. The Second scenario tested the possibility of maximizing the energy efficiency by using replacement materials, while keeping the original wall thickness. A remarkably better energy performance was achieved with the use of thermal insulation with ordinary construction materials. The positive impact of the use of thermal insulation on the energy performance levels has several implications:

- In cases where the same-type material resources are not available, the heritage structures can still be reconstructed on the same setting, following the same design, but with replacement materials. In such cases, the wall thicknesses may or may not be the same as the original dimensions but, while the material authenticity will be lost, the energy performance will be maximized.
- Existing stones can be used to reconstruct parts of the walls and the rest can be reconstructed with replacement materials and/or material layers.
- The energy performance can be further improved by adding thermal insulation layers on the roofs, which are reconstructed with same-type materials.
- Although adding thermal insulation layers in the building envelope/fabric would increase the reconstruction costs, the operational costs would be significantly reduced as well as the carbon-dioxide emissions, and for this reason, such costs would therefore be justified.
- The walls can be made thinner using a lesser amount of stone, particularly in the cases where the main source for stone is salvaging from other demolished structures. With added thermal insulation layers, the walls will have the originally designed thickness with better energy performance. However, further research is required to determine the types of thermal insulation layers to ensure that they are physically and chemically compatible with coral stone walls.
- In general, effective water proofing is required in future reconstructions to prevent water absorption as, depending on the type of insulation material used, the moisture content could increase in the building envelope/fabric (i.e., walls and even roofs).

With regard to the hygrothermal response of the thermal conductivity of the insulation material used and its impact on a building's energy performance, there has been some credible research where the findings showed results between 2–10%, depending on the type of insulation material. In this case, the moisture content increased significantly—for example with the rockwool type of insulation. While this is understandable, for other types of material, such as EPS insulation, the moisture content showed very little difference under conditioned building environments and when assessing the impact of the thermal conductivity under constant and variable temperature with different levels of moisture content [40–43].

Future research in this field will help in developing materials and formulating material layers suitable/acceptable for reconstructions to further reduce the primary energy consumption, in particular the cooling loads, hence, contributing to the reduction of $CO_2$ emissions.

However, as Dewi [15] demonstrated elsewhere, reconstruction with different materials is not yet an officially accepted process, causing the removal of the concerned heritage structures from the official local and global heritage lists, including the WH List. Therefore, despite the comparable advantages in reducing the cooling loads, as was shown in our simulations, thermal insulation is not currently used in heritage reconstructions in Kuwait [30] or elsewhere in the Gulf region [36] as there is a consensus in following the principles of the Venice Charter.

The heritage authorities, rightfully so, do not want to jeopardize the chances of their historic structures to be recognized as WH Sites. Yet, while the Venice Charter [6] and subsequent doctrinal documents continues to provide guidance in many contexts, as other authors have noted [16], the theoretical foundations must continue evolving to be more inclusive and responsive. It is important that the guidance and goals pursued in reconstruction planning in post-conflict or post-disaster contexts should have relevance to the local cultural and environmental realities. The possibility of future increases in the currently small proportion of community members in Kuwait who indicated no interest in the heritagization of the Diwan, must be carefully considered in the context of the globalization of local architectural heritage. This may happen if the fabric authenticity criteria continue to dominate the reconstruction planning and future generations start to perceive the physical preservation of heritage structures as environmental burdens.

## 6. Conclusions

While there is a consensus on the cultural and psychological reasons why communities might wish to reconstruct the historic buildings in the aftermath of catastrophes, what is appropriate and not appropriate in treating the material fabric is debated. Authorized heritage discourses still favor material authenticity. Debates and disagreements extend the decision-making processes, resulting in further decay of the existing materials and architectural elements. On the other hand, the decreasing natural material resources call for special emphasis to be placed on environmental protection as part of heritage reconstruction planning and strategies.

This is a growing problematic area, which must be addressed and negotiated while shaping the policies of heritage reconstruction both at the national and supra-national levels. It is important that those in policy- and decision-making processes are more open and receptive to newly emerging perspectives on environmental concerns and realities, such as material resources and energy efficiency. Put differently, this is a key justification for the development of inclusive and responsive reconstruction methodologies and policies. Material authenticity must be re-defined and re-positioned within the context of historicist reconstruction, considering the dilemma when cultural and environmental sustainability goals and fabric-authenticity concerns are on the opposite positions.

With the increased heritage destruction due to wars in the MENA region and elsewhere, as well as devastating natural catastrophes, the nature and the role of replacement materials needs to be reconsidered. Heritage reconstructions, while re-establishing the broken link for cultural continuity, should also respond to the unprecedented climatic change and diminishing natural material resources. How local governments and global heritage authorities perceive their role in negotiating for environmental protection and the growing threat of climate change in the creation of sustainable architectural heritage, and how this translates into new forms and processes of heritage, remains to be seen.

**Author Contributions:** Conceptualization, R.S., D.A. and N.A.; methodology, R.S., H.A., D.A. and N.A.; software, validation, H.A. and D.A.; investigation, resources, R.S., D.A., and N.A.; formal analysis, R.S., and H.A.; writing—original draft preparation, R.S., and H.A.; visualization, H.A., D.A. and N.A. All authors have read and agreed to the published version of the manuscript.

**Funding:** This research received no external funding.

**Conflicts of Interest:** The authors declare no conflict of interest.

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
