# Peer review of "Heritage Reconstruction Planning, Sustainability Dimensions, and the Case of the Khaz’al Diwan in Kuwait"

_sustainability, doi:10.3390/su12218805_

Round 1
Reviewer 1 Report
I really enjoyed reading this paper. I think that it brings up some important points regarding the politics of reconstruction, particularly when building materials are environmentally protected.
A couple of things to strengthen the paper.
- There are some minor English edits that need to be made throughout the paper.
- p. 1, line 42. What is meant by "doctrinal"?
- The authors refer to events, conferences, and documents that assume that the reader has prior knowledge of these things. For example, what was the Nara document?? What did it contact? The authors need to tell readers not just that these things are important, but what they were.
- I was unclear in the first three paragraphs whether these standards were with regards to world heritage sites.
- p. 2, line 67: What are these disappointing results?
- p. 2, line 68: Expand on that is "appropriate and inappropriate".
- p. 2, lines 91-93. Incomplete sentence.
- p. 6: What is the Venice Charter?
- p. 6: What is meant by a site needs to be environmentally and culturally sustainable.
- p. 8, lines 278-279. The authors say that there has been wide attention on energy saving solutions, but list only one reference. The authors need to list 2-3 more references here.
- p. 11, lines 333-334: D What do "spiritual values" have to do with the case study Explain.
Author Response
Responses to Reviewer-1’s feedback
Reviewer 1: I really enjoyed reading this paper. I think that it brings up some important points regarding the politics of reconstruction, particularly when building materials are environmentally protected. A couple of things to strengthen the paper:
We would like to thank the reviewer as their comments have strengthened the paper. Revised parts are shown in the manuscript in ret font. Our responses are as follows:
1) There are some minor English edits that need to be made throughout the paper.
Response: We are submitting the paper to the Journal’s proofreading/editing service.
2) p. 1, line 42. What is meant by "doctrinal"?
Response: While ‘doctrinal’ is used for international documents which codify international conservation principles, we decided to remove it and edit the text as follows:
‘Hence, documents aiming for international conservation standards include rather rigid definitions for the conditions that justifies reconstruction, cautiously restricting it to exceptional circumstances such as destructions due to wars or natural catastrophes.’
3) The authors refer to events, conferences, and documents that assume that the reader has prior knowledge of these things. For example, what was the Nara document?? What did it contact? The authors need to tell readers not just that these things are important, but what they were.
Response: We included information in the text regarding the referred international documents.
4) I was unclear in the first three paragraphs whether these standards were with regards to world heritage sites.
Response: We edited the text in the first three paragraphs to make it clearer that these are the general international conservation standards, and the WH’s stance and position in relation to these standards.
5) p. 2, line 67: What are these disappointing results?
Response: While the authors of the source we cited has mentioned the ‘disappointing results’ without exemplifying them, reading their text, it is clear that these were about use of un-scientific and mixed methods in reconstructions. So, we deleted ‘disappointing results’ from the text and revised the sentence as follows:
In other cases, haphazardly planned rapid reconstructions immediately after disasters have resulted in incoherent and controversial treatment of historic fabric [16].
6) p. 2, line 68: Expand on that is "appropriate and inappropriate".
Response: We revised the text as follows:
Other authors argue that there is no sufficient information to determine what are the appropriate or inappropriate methods in reconstruction planning in general [13],
7) p. 2, lines 91-93. Incomplete sentence.
Response: We revised the text as follows:
‘Cultural continuity’ is addressed as the main pillar underpinning architectural conservation processes, and ‘environmental protection’ is addressed in relation to the use of construction materials/layers and their impact on the environment.
8) p. 6: What is the Venice Charter?
Response: The text has been revised and Venice Charter is explained in the second paragraph of the manuscript as follows:
Venice Charter of 1964, which has been the first influential document to codify internationally accepted architectural conservation standards, has prioritized materialist authenticity, and ruled out all reconstruction work ‘a priori’ [6].
9) p. 6: What is meant by a site needs to be environmentally and culturally sustainable.
Response: The text has been revised as follows:
The clause 5.5.8 of the document states that the proposed use of the site should be ‘environmentally and culturally sustainable’, in conformity with WH criteria [28]. While the clause does not define the ‘environmental and cultural sustainability’, it sets forth a few generic goals. Accordingly, ‘cultural sustainability,’ is associated with contribution to the ‘quality of life of communities concerned’, and ‘environmental sustainability’ is associated with having ‘a minimum environmental impact in design, construction and operation phases.’ It is also highlighted in the same clause that such sustainable use or any other change should not impact adversely on the OUV of the site. Put differently, authenticity of the fabric should not be compromised by the set sustainability goals.
10) p. 8, lines 278-279. The authors say that there has been wide attention on energy saving solutions, but list only one reference. The authors need to list 2-3 more references here.
Response: More references have been added.
11) p. 11, lines 333-334: D What do "spiritual values" have to do with the case study Explain.
Response: We quoted the full sentence from the cited document, and actually ‘spiritual values’ do not relate to our case study. So, we deleted ‘spiritual values’ and revised the sentence as follows:
It has been almost four decades since the Declaration of Dresden justified how the desire to acknowledge architectural monuments both intellectually and politically were among the key reasons for initiating their reconstruction [3].

Reviewer 2 Report
This paper presents a most interesting perspective by examining the thermal benefits of reconstructing of this attraction either with authentic materials or with newer materials. While it is expected that the appearance of the building would be very similar to that of the original structure.
While the paper presents an excellent perspective, it would have been useful to present a more comprehensive assessment of the total restoration and operating costs under both scenarios. When compared with potential revenues the paper might have been able to address the full scope of sustainability. The entire economic sustainability from a comprehensive perspective by producing pro forma revenue and expense statements along with an economic impact assessment.
The social-cultural sustainability, some of the elements of which were assessed in the survey and in the environmental sustainability. Given the impacts of the weather conditions on the existing structure it would be useful to incorporate this element into the assessment. Going beyond the energy efficiency to include assess the economic life of the site given the environmental conditions and any estimates of the annual costs allocated for maintenance.
Overall this is a very good paper, well written. However it falls short in its assessment of the project's sustainability.
Author Response
Responses to Reviewer-2’s feedback
Reviewer: This paper presents a most interesting perspective by examining the thermal benefits of reconstructing of this attraction either with authentic materials or with newer materials. While it is expected that the appearance of the building would be very similar to that of the original structure.
The authors would like to thank Reviewer 2 for their constructive feedback.
While the paper presents an excellent perspective, it would have been useful to present a more comprehensive assessment of the total restoration and operating costs under both scenarios. When compared with potential revenues the paper might have been able to address the full scope of sustainability. The entire economic sustainability from a comprehensive perspective by producing pro forma revenue and expense statements along with an economic impact assessment.
Response: We have revised section 4 including further analysis containing total restoration and operating costs under both scenarios as well as have included carbon dioxide emissions to further demonstrate environmental impact of both scenarios.
The social-cultural sustainability, some of the elements of which were assessed in the survey and in the environmental sustainability. Given the impacts of the weather conditions on the existing structure it would be useful to incorporate this element into the assessment. Going beyond the energy efficiency to include assess the economic life of the site given the environmental conditions and any estimates of the annual costs allocated for maintenance.
Response: We have considered the impacts of the weather conditions on the existing structure and this has already been factored in the calculations performed through the building simulation software and analysis. Climatic data has been incorporated into all calculations using meteorological data from the Kuwait International Airport. In addition, we have looked at the hydrothermal issues where the moisture content may have some impact on thermal conductivity of the walls and roof components, and for this reason, we have now included some discussions and further recommendations to avoid such likely issues depending on the type of insulation material use for reconstruction.
Overall this is a very good paper, well written. However, it falls short in its assessment of the project's sustainability.
Response: We have now revised our paper covering further assessment on unit costs for reconstruction as well as annual carbon-dioxide emissions due operational energy under both scenarios.
